# Alternative mRNA Processing of Innate Response Pathways in Respiratory Syncytial Virus (RSV) Infection

**DOI:** 10.3390/v13020218

**Published:** 2021-01-31

**Authors:** Xiaofang Xu, Morgan Mann, Dianhua Qiao, Allan R. Brasier

**Affiliations:** 1Department of Medicine, University of Wisconsin-Madison School of Medicine and Public Health (SMPH), Madison, WI 53705, USA; xxu@wisc.edu (X.X.); mwmann@wisc.edu (M.M.); dqiao@wisc.edu (D.Q.); 2Institute for Clinical and Translational Research (ICTR), University of Wisconsin-Madison, Madison, WI 53705, USA

**Keywords:** alternative splicing, differential polyadenylation, interferon regulatory factor (IRF), interferon (IFN), Iso-Seq™, single-molecule, real-time (SMRT)

## Abstract

The innate immune response (IIR) involves rapid genomic expression of protective interferons (IFNs) and inflammatory cytokines triggered by intracellular viral replication. Although the transcriptional control of the innate pathway is known in substantial detail, little is understood about the complexity of alternative splicing (AS) and alternative polyadenylation (APA) of mRNAs underlying the cellular IIR. In this study, we applied single-molecule, real-time (SMRT) sequencing with mRNA quantitation using short-read mRNA sequencing to characterize changes in mRNA processing in the epithelial response to respiratory syncytial virus (RSV) replication. Mock or RSV-infected human small-airway epithelial cells (hSAECs) were profiled using SMRT sequencing and the curated transcriptome analyzed by structural and quality annotation of novel transcript isoforms (SQANTI). We identified 113,082 unique isoforms; 28,561 represented full splice matches, and 45% of genes expressed six or greater AS mRNA isoforms. Identification of differentially expressed AS isoforms was accomplished by mapping a short-read RNA sequencing expression matrix to the curated transcriptome, and 905 transcripts underwent differential polyadenylation site analysis enriched in protein secretion, translation, and mRNA degradation. We focused on 355 genes showing differential isoform utilization (DIU), indicating where a new AS isoform becomes a major fraction of mRNA isoforms expressed. In pathway and network enrichment analyses, we observed that DIU transcripts are substantially enriched in cell cycle control and IIR pathways. Interestingly, the RelA/IRF7 innate regulators showed substantial DIU where major transcripts included distinct isoforms with exon occlusion, intron inclusion, and alternative transcription start site utilization. We validated the presence of RelA and IRF7 AS isoforms as well as their induction by RSV using eight isoform-specific RT-PCR assays. These isoforms were identified in both immortalized and primary small-airway epithelial cells. We concluded that the cell cycle and IIR are differentially spliced in response to RSV. These data indicate that substantial post-transcriptional complexity regulates the antiviral response.

## 1. Introduction

Respiratory syncytial virus (RSV) is responsible for seasonal outbreaks of respiratory tract infections worldwide. This enveloped, single-stranded, negative-sense RNA virus of the human *Orthopneumovirus* genus of the Pneuomoviridae family is the most common cause of pediatric hospitalization in children less than 5 years of age [1]. After inoculation into the nasopharynx, RSV fuses with ciliated epithelial cells and replicates to high levels, spreading throughout the airways to the conductive and lower-airway cells involved in gas exchange. Here, productive infection in small-airway epithelial cells induces the rapid secretion of cytokines [2], interferons [3], exosomes [4], and damage-associated patterns [5] that mediate mucus production and leukocytic inflammation. The findings that the initial clinical manifestations of hypoxia are correlated with epithelial-derived cytokines [6,7] detected in the airway fluids of naturally infected children and experimentally challenged adults [7,8,9] indicate that the innate immune response (IIR) plays an important role in the pathogenesis of disease.

Airway epithelial cells sense viral replication through a network of pattern recognition receptors (PRRs) monitoring the airway lumen, cellular cytoplasm, and subcellular organelles for the presence of pathogen-associated molecular patterns (PAMPs, [10,11]). Double-stranded RNA (dsRNA) and 5-phosphorylated RNA are viral PAMPs generated by productive RSV replication that bind the intracellular retinoic-acid-inducible gene-I (RIG-I). Activated RIG-I binds the mitochondrial antiviral signaling protein (MAVS) to form an intracellular signaling complex on the mitochondrial surface, known as the inflammasome. The inflammasome is a signaling receptor scaffold that recruits and activates kinases and ubiquitin ligases that are rate limiting in the activation of the interferon (IFN) regulatory factor (IRF)3 and nuclear factor-κB (NFκB) transcription factors, stimulating their nuclear translocation [12].

The nature, timing, and magnitude of inducible epithelial cytokine responses play important roles in shaping disease and activating adaptive immunity. In the epithelium, IRF and NFκB crosstalk is the key regulator of type I and type III IFN gene expression [13,14], responsible for the expression of some 350 interferon-simulating genes (ISGs), producing antiviral protection. IFN type I/III stimulation produces a dramatic upregulation of IRF, TLR3, and RIG-I to further amplify the antiviral response. This IFN amplification loop primes neighboring epithelial cells to elicit a protective antiviral state and commitment to apoptosis [15].

The genomic response of RSV-infected airway epithelial cells has been studied systematically using high-density microarrays [2,16] and short-read RNA sequencing [17] to identify time-dependent, replication-sensitive [18], and NFκB-controlled gene expression programs [16,19]. Although these studies have provided information at the gene expression level, mRNA-processing events, including alternative splicing (AS) and alternative polyadenylation (APA), are recognized to generate transcriptome complexity and proteome diversity. These events have been underexplored due to technical limitations in full-length mRNA sequencing. Recent advances in high-throughput sequencing of full-length transcripts now enables the identification of differentially processed mRNAs and opens the door for understanding transcriptome diversity in normal and cellular stress responses.

In this manuscript, we use single-molecule, real-time (SMRT) RNA sequencing to explore regulated alternative mRNA processing in a human small-airway epithelial cell (hSAEC) model in response to RSV replication. SMRT sequencing data were subjected to a pre-processing informatics pipeline, yielding high-confidence long-read transcripts that characterize the composition of the full-length transcriptome. Differentially expressed isoforms were identified using an expression matrix from short-read RNA sequencing data, resulting in the identification of 905 genes undergoing APA and 355 genes expressing differential isoform utilization (DIU). DIU isoforms represent a switch in the major mRNA isoform encoded by a gene. We found that DIU genes are enriched in cell cycle control and IIR pathways. We focused on validating master transcriptional regulators of the IIR, RelA, and the inducible IRF isoform, IRF7, critical in the control of the type I and III IFN response. Differential processing was confirmed in primary cultures of human small-airway cells. We concluded that the IIR is regulated extensively by AS.

## 2. Materials and Methods

### 2.1. Cell Culture

Human small-airway epithelial cells (hSAECs) from a cadaveric donor were immortalized using human telomerase/CDK4, as previously described [20,21]. These non-oncogenic, telomerase-immortalized cells maintain genomic and proteomic signatures representative of primary SAECs over many population doublings [21] and manifest characteristic cell-state transitions typical of primary hSAECs [20,22,23]. hSAECs were grown in small-airway epithelial cell growth medium (Lonza, cc-3118) in a humidified atmosphere of 5% CO_2_. For validation experiments, primary small-airway epithelial cells (pSAECs) were obtained from cadaveric donors (ATCC, PCS-301-010). Cells were grown in airway epithelial cell basal medium (ATCC, PCS-300-030) supplemented with a bronchial epithelial cell growth kit (PCS-300-040).

### 2.2. Virus Preparation and Infection

The human RSV long strain was grown in Hep-2 cells and prepared, as previously described [24,25]. The viral titer of purified RSV pools was varied from 8 to 9 log PFU/mL, determined by a methylcellulose plaque assay [26]. Viral pools were aliquoted, quick-frozen on dry ice–ethanol, and stored at −70 °C until they were used. The virus was adsorbed onto the surface of hSAECs in submerged culture for 2 h prior to washing at an multiplicity of infection (MOI) of 1. Mock infected cells were used as controls.

### 2.3. Isoform-Selective RT-PCR

Cells were harvested for RNA isolation using an RNeasy kit with on-column DNase digestion (Qiagen). Synthesis of complementary DNAs (cDNAs) was done using a First Strand cDNA Synthesis Kit (Thermo Scientific). Semi-quantitative reverse transcriptase PCR (RT-PCR) assays to detect the presence of 8 alternative exons were developed using gene- and exon-specific PCR primers. Sequences, location of hybridization, and anticipated sizes of tested AS events are shown in Table 1. After quantitation by SYBR Green, PCR products were size-fractionated in 2% agarose gel. The gel contained GelRed dye (10,000× dilution) from Biotium (41003). The ladder was a 1 kb plus DNA ladder from Invitrogen (10787018). After photography, DNA bands were extracted using a Qiagen kit (28706). Purified DNA was subjected to Sanger sequencing for splice site validation.

### 2.4. SMRT Sequencing

RNA isolation was performed as above; the same RNA was used to generate SMRTbell libraries and prepare mRNAs for short-read sequencing. For SMRT sequencing, Iso-Seq™ SMRTbell libraries containing circular templates were generated using an NEBNext Single Cell RT Enzyme Mix and recommendations by the manufacturers for long-read transcripts using the Iso-Seq™ Express Template (PN: 101-901-700) and indicated quality control (QC) steps. Sequencing was performed using the Pac Bio Sequel II system with SMRT Cell 8M.

### 2.5. Short-Read RNA Sequencing

Libraries were produced using the TruSeq Stranded mRNA and subjected to Illumina HiSeq 2000 paired-end sequencing. The trimming software skewer was used to process raw fastq files and QC statistics. The trimmed paired-end reads were aligned against human genome hg38 using Salmon 1.3 [27]. Mapped paired-end reads for both genes and transcripts (isoforms) were counted in each sample using RNA-Seq by Expectation Maximization (RSEM). Contrasts were compared for genotypes and treatment conditions using DESeq2 [28].

### 2.6. Data Analysis

Identification, polishing, and annotation of the transcriptome from SMRT sequencing was performed using the PacBio Iso-Seq™ bioinformatics pipeline [29]. Polished, annotated genes were analyzed by SQANTI [30]. TappAS was used to identify functional annotations [31].

## 3. Results

To investigate the effect of RSV infection on mRNA processing, we conducted short-read and SMRT RNA sequencing in a well-characterized model of telomerase-immortalized *Scgb1a1*+ human small-airway epithelial cells (hSAECs). This model was selected because RSV replicates in the small airways in lung infection [8], and these cells demonstrate replication-dependent pathogenic mucin- and T helper 2 lymphocyte-activating cytokines that mediate disease [4]. hSAECs were infected with sucrose-cushion-purified RSV (Long, MOI = 1) for 0 or 24 h using *n* = 4 independent biological replicates. Short-read sequencing and SMRT RNA sequencing were analyzed according to the data analysis pipeline in Figure 1.

### 3.1. Annotation of hSAEC Transcriptome

The structural and quality annotation of novel transcript isoforms (SQANTI) analysis pipeline was used for a systematic characterization and curation of the full-length transcriptome [30]. In the curated transcriptome, 113,082 unique isoforms were identified, encoded by 13,038 genes (12,013 annotated and 1025 novel). Characterization of the transcripts by canonical splice junctions included 28,561 represented full splice matches (FSMs), and 35,495 were incomplete splice matches (ISM) (Table 2). For the purposes of this analysis, canonical splice junctions correspond to those with the dinucleotide GT at the beginning of an intron, followed by AG at the end, as well as GC-AG and AT-AC; these combinations are present in more than 99.9% of all human introns [30,32].

The length distribution of the curated hSAEC transcriptome had a median of approximately 1 kB in length with an asymmetric tail including transcripts of up to 10 kB (Figure 2A). Multi-exon genes constituted the genes with the largest size; single-exon genes were substantially smaller (Figure 2A). Interestingly, over 45% of genes encoded six or greater mRNA isoforms (Figure 2B), with the majority of these genes corresponding to annotated genes (Figure 2C). By contrast, novel genes were skewed toward a single isoform (Figure 2C). During SMARTbell library synthesis, reverse transcriptase (RT) can jump across RNA secondary structures, producing artifacts in the interpretation of splicing events. SQANTI identifies these artifacts by an algorithm identifying RNA direct repeats upstream of a noncanonical intron [30,32]. Consequently, 0% of canonical and 3% of novel transcripts with canonical splice junctions show evidence of RT switching (Figure 2D). By contrast, 9% of novel genes with noncanonical splice junctions were likely due to RT switching and therefore excluded from subsequent analysis. We further noted that >60% of transcripts were mapped to ±50 nt of annotated transcription start sites (TSS; Figure 2E). Similarly, ~45% of polyadenylation (polyA) motifs were found within ±500 nt of the 3′ end of the gene. The median location was within 18 nt of the 3′ end of the gene (Figure 2F). The median length and feature distribution of the hSAEC transcriptome is consistent with other reports of mammalian transcriptomes [33,34].

### 3.2. Alternative Polyadenylation (APA) Site Analysis

SMRT sequencing provides genome-wide information about changes in APA site utilization (APAU) in response to RSV infection. In this study, 905 genes had statistically significant changes in APAU in response to RSV; of these, 108 transitioned to distal polyA site utilization, and 707 exhibited proximal polyA site utilization. Figure 3A shows a volcano plot of polyadenylation site switching between proximal and distal 3′ untranslated tract (UTR) polyadenylation sites. In this analysis, a positive APAU fraction corresponds to transcripts with distal APA usage in RSV infection, whereas a negative APAU indicates the proximal APA site is utilized in RSV-infected cells. The filtered APAU genes (>5% APAU) were next subjected to pathway analysis. Individual identities, expression values, and analysis metrics for APAU genes are given in Appendix A.

Pathway enrichment of APAU genes was conducted in the Reactome Pathway Knowledgebase [35], a manually curated and peer-reviewed relational database. Strikingly, APAU-gene-populated pathways were highly enriched in ribosomal proteins important in signal recognition peptide (SRP) protein targeting, Cap-dependent protein translation, and nonsense-mediated mRNA decay (Figure 3B). Individual identities and enrichment data for the top pathways are given in Appendix A The individual genes included numerous members of the ribosomal protein L (RPL) family (RPL-2, 6, 7, 8, 9, 13, 14, 15, 17, 22, 23, and 32), ribosomal S protein (RPS-A, 4X, 15A), and eukaryotic translation initiation factor (EIF4A) (Appendix A). It is noteworthy that RSV-infected cells do not exhibit the host cell translational shut-off characteristic of other RNA virus infections [36].

### 3.3. Differential Expression of Alternatively Spliced mRNA Isoforms Using Functional Iso-Transcriptomics (FIT) Analysis

To identify differentially expressed transcripts, quantitation of mRNA isoforms in the curated transcriptome was first performed using short-read expression in DESeq2 using independent hypothesis weighting [28]. This information, including the experimental design matrix and the genome annotation file from the curated transcriptome characterized previously, was subjected to functional isotranscriptomics (FIT) analysis in tappAS (Figure 1) [31].

Principal component analysis (PCA) was used to determine variability among replicates and the global effect of RSV on gene expression. Here, PCA indicated that over 80% of the variance in the data sets was as a result of the RSV Infection (Figure 4A) and that the replicates closely grouped together. The RSV-infected replicates showed slightly more variability, accounting for 4% of the variation, indicating the dynamic effects of RSV on gene expression (Figure 4A). Collectively, we interpreted this analysis to indicate that the data were reproducible in the biological replicates, and proceeded with differential analysis, in which 4286 genes were significantly changed by RSV infection (*p* < 0.05, using multiple hypothesis correction in DESeq2). Of these, a larger number of genes were downregulated in response to RSV infection (Figure 4B), a finding consistent with earlier high-density cDNA microarray studies [2]. To focus on abundant genes that were most significantly changed, mRNA isoforms in the curated transcriptome were filtered by 1.5 log2 fold change (RSV vs. control).

Approximately 50% of mRNA transcripts showed some degree of differential isoform utilization (DIU) as a function of RSV infection (Figure 4C and scatterplot in Figure 4D); of these, 355 genes were identified to have significant isoform switching. DIU identifies genes that have distinct splice forms as a function of RSV infection that represent the majority of the isoforms expressed. Differentially expressed genes were enriched in coding sequences (CDS) and post-translational modification (PTM) acceptor sites and relatively depleted in miRNA-binding sites relative to all features in the transcriptome (Figure 4E).

### 3.4. Functional Analysis

DIU genes were analyzed for pathway over-representation (Figure 5). From this analysis, we observed that DIU transcripts were significantly enriched in the cell cycle, innate signaling, and chromatid separation in mitosis pathways (Figure 5A). The same set of DIU genes was subjected to pathway enrichment analysis in the Reactome Knowledgebase. Over 15 highly significant pathways were identified; the top 10 over-represented pathways are shown in Figure 5B. Of these, endosomal processing, antigen/major histocompatibility complex (MHC) class I antigen processing, and IFN cytokine signaling were identified as having a significant fraction of the pathway represented. The pathway enrichment and genes mapping to them are shown in Appendix A. We noted that the interferon-stimulated genes, including MX Dynamin-Like GTPase 1 (MX1/2), HLA-B,C, IFN-Induced Protein with Tetratricopeptide Repeats (IFIT1,2,3), 2′-5′-Oligoadenylate Synthetase 1 (OAS)-L/3, Interferon-Stimulated Exonuclease Gene 20 (ISG20), and interferon regulatory factor (IRF)-1 and IRF7 were major genes constituting these pathways. The individual genes and values are given in Appendix A. We noted that both analyses showed substantial signals of cell cycle regulation and IFN signaling.

### 3.5. Network Analysis

Differentially expressed DIU transcripts were next analyzed for network representation, incorporating protein–protein interaction information. In a result that was consistent with the biological pathway and network enrichment analyses (Figure 5), four major networks were identified. These included cell cycle checkpoint, metaphase/anaphase, IFN signaling, and IFN α/β signaling (Figure 6). Noted in the pathway enrichment earlier, a number of well-established antiviral IFNs and IFN regulators are seen, including IFN lambda (IFNL1), OAS, MX1, IFIT1, and IRF-1 and IRF-7 (Figure 6). To elaborate, IFNL1/IL29 is an important mucosal antiviral and pro-apoptotic IFN [37]. OAS activates latent RNase L, which results in viral RNA degradation and reduces viral replication [38]. MX1 encodes a guanosine-triphosphate-metabolizing protein that participates in the cellular antiviral response [39]. IRF-1 and IRF-7 are NFκB-inducible transcription factors in epithelial cells that cooperate in the regulation of type I and III IFN production [40]. Taken together, the pathway and network analysis consistently indicate that the IFN pathway undergoes substantial DIU in response to RSV infection.

### 3.6. Isoform Characterization

To understand the patterns of AS in the antiviral response, the DIU transcripts for interferon regulatory factor (*IRF*)*-1/7* and the 65 kDa NFκB transcriptional subunit (*RelA*) were extracted and SMRT isoforms were compared to the exon abundances of each transcript determined by short-read RNA sequencing analysis. For each gene, the full splice match (FSM) transcripts were extracted and compared to exon abundance in the integrated genome viewer (IGV) in Figure 7. Exon abundance quantified by short-read RNA sequencing analysis is represented in the pile-up graph in the top panel of each gene. *IRF7* shows marked alternative TSS utilization, as well as intron inclusion in introns 9–10 (Figure 7A). Notably, a number of isoforms lack the NH2 terminal IRF DNA-binding domain. We identified 22 *RelA* isoforms by SMRT sequencing; these variants are encoded by alternative TSS utilization, intron inclusion, and exon skipping (Figure 7B).

### 3.7. Annotation of RelA and IRF7 AS Isoforms

To understand the functional implications of RelA and IRF7 alternative mRNA processing, we conducted a functional isoform analysis in tappAS [31]. Here, the major DIU isoforms are displayed for coding and noncoding mRNA transcripts with the PacBio sequence identifiers. Where possible, each is mapped to previously known (RefSeq/NCBI-curated) transcript variants or, where no annotation exists, labeled as novel splice forms. Of the five RelA isoforms identified in our analysis, three were curated and two were novel DIU isoforms (Figure 8A); only four were significantly expressed relative to the wild-type RelA mRNA. The predicted translation of PB6969.5 encodes a 520 aa peptide aligning to aa 32-551 of the canonical RelA protein; PB6969.13 encodes a polycistronic mRNA, with the upstream coding sequence identical to the 520 aa RelA protein and a 71 aa in-frame peptide downstream with a consensus initiation methionine residue. This downstream peptide has no distinguishing protein domains. PB6969.12 encodes three in-frame polypeptides with initiator methionines; the second and largest peptide is 186 aa (18.8 kDa), homologous to RelA transcript variant 2.

Similarly, seven IRF7 isoforms were identified, and splicing patterns of the coding and noncoding transcripts are shown in Figure 9. Of these, five were novel AS forms and two were extant in the NCBI database (Figure 9A). Out of the seven isoforms identified, four DIUs were as abundant as the wild-type IRF7 mRNA (Figure 9B). PB.6624.20 and PB123733 isoforms encode COOH terminal truncations of IRF7 and lack the NH2 terminal IRF DNA-binding domain.

### 3.8. RelA Isoform Validation

The annotated RelA gene in the hg38 assembly includes 12 exons (Figure 8A). Four RefSeq annotated mRNAs have been deposited and validated; these variants include alternative splicing in exons 11 and 12. Our data extends the understanding of RelA isoforms due to the inclusion of introns 1 and 8 in distinct AS events (Figure 8A). Three of the most abundant RelA isoforms (PB.6969.12, PB.6969.13, and PB.6969.5) were selected for independent validation. For this, a qualitative RT-PCR approach was developed using gene- and exon-flanking primer sets, diagrammed in Figure 10, and sequences presented in Table 1. hSAECs were infected in the absence (mock) or presence of pRSV (MOI = 1), and total RNA was subjected to isoform-specific RT-PCR. The PCR products were size-resolved by agarose gel electrophoresis (Figure 10B). Notably, primer set #1 detected a major ~270 nt fragment and a less abundant 96 nt fragment, confirming the presence of occult exons contained in the first intron (Figure 10A). To validate these splice forms, gel bands were excised and sequenced independently, confirming the presence of each. For primer set #2, representing the inclusion of an exon within intron 8, a 205 nt fragment was produced. A longer 876 nt fragment was not produced, so the transcripts with inclusion of intron 11 could not be validated.

We sought next to determine whether novel isoforms were RSV-inducible. For this purpose, quantitative real-time PCR (qRT-PCR) Sybr Green assays were developed. For each isoform, we determined the fold change of RSV-infected cells vs. mock treatment (Figure 10C). Of the isoforms assayed, we noted that PB6969.13 (containing intron 1) is the most highly inducible by RSV infection, showing a 13-fold induction (*p* < 0.01, *t*-tailed *t*-test). Note that all isoforms are induced by RSV infection, consistent with their earlier definitions as DIU isoforms.

### 3.9. IRF7 Isoform Validation

The annotated IRF7 gene includes 10 exons (Figure 9A). Three variants have been annotated in RefSeq, corresponding to alternative splicing on exon 3 and occlusion of exon 7. Four of the most abundant IRF7 isoforms (PB.6624.7, PB123733, PB6624.10, and PB6624.20) were selected for validation, with primer sets and expected sequences shown in Figure 11 and Table 1. Qualitative RT-PCR assays were performed on RSV-infected hSAECs, as above, and fractionated by agarose gel electrophoresis. Primer set #4 detected an 82 nt fragment corresponding to the presence of an exon contained within intron 4 in PB6624.7 (Figure 11B). However, we were not able to detect the inclusion of intron 6 using primer set #5. This discrepancy may be due to limitations in assigning short-read RNA sequencing counts to SMRT data or qualitative PCR optimization. Primer set #6 produced a 400 nt fragment, indicating the inclusion of exons 4–6. Primer set #7 produced two fragments, indicating the presence of exon occlusion within annotated exon 1. Finally, primer set #8 produced a 111 nt fragment corresponding to the splicing of intron 8 and a lower abundant 199 nt fragment corresponding to the inclusion of intron 8 in PB6624.10.

To determine the effect of RSV infection on IRF7 isoform expression, quantitative real-time PCR (qRT-PCR) Sybr Green assays were developed and used to determine the fold change in RSV-infected cells vs. mock-treated cells (Figure 11C). RSV induced te expression of all the IRF7 primer pairs.

Finally, we determined whether AS splice events were similar to those seen in primary human small-airway epithelial cells. For this purpose, cytokeratin-19-positive primary human pSAECs isolated from the small airways of cadaveric donors were cultured and then infected in the presence or absence of RSV, and RNA was extracted. We observed an identical splice pattern in qualitative RT-PCR for primer pairs 4–7 (Figure 11D). A similar pattern of RSV inducibility was also observed in the pSAECs compared to the telomerase-immortalized hSAECs (compare Figure 11E,C).

## 4. Discussion

RSV is a major human pathogen that represents the most common cause of pediatric hospitalization in young children. Because RSV replicates to high levels in airway epithelial cells, where the timing and character of the IIR epithelial cells plays an important role in the pathogenesis of disease and resolution, understanding the IIR to RSV in models relevant to disease is important. Previous work has shown that small-airway epithelial cells rapidly secrete cytokines [2], IFNs [3], exosomes [4], and damage-associated patterns [5] in response to RSV replication. In this study, we applied single-molecule real-time (SMRT) RNA sequencing to explore regulated AS in a human small-airway epithelial cell (hSAEC) model in response to RSV replication. Using an established informatics pipeline, we identified substantial numbers of mRNA-processing events in our curated transcriptome of 113,082 unique transcripts encoded by 13,038 genes. These events include alternative polyadenylation utilization, cryptic exonic inclusion, alternative splice site utilization, exon occlusion/skipping, and intronic inclusion. To understand the role of mRNA processing in the IIR, we focused on the detailed AS events of differentially expressed mRNAs exhibiting differential inclusion utilization (DIU), a designation indicating the abundance of major splice forms are changed by RSV infection. Viewed through this lens, we identified 355 mRNA transcripts that were induced by 1.5 log2 fold change with a false discovery rate (FDR) of 0.05 by RSV replication and exhibited DIU. RSV-regulated DIU transcripts included AS events with inclusion of exons that encode different miRNA-binding sites and alternative post-translational modifications. Functional genome ontology (GO) enrichment and network analysis shows the DIU genes are highly represented in cell cycle regulation and major transcriptional regulators of the IIR.

### 4.1. SMRT Read Technology as a High-Resolution Method for Identifying AS and APA Regulation

Although it is assumed that post-transcriptional mRNA processing represents a mechanism for substantial transcriptome and proteome diversity, the ability to directly identify and quantify changes in AS and APA have not been previously possible using conventional short-read mRNA. It is well established that short-read RNA sequencing lacks sufficient sensitivity and read length to reliably reconstruct individual AS isoforms [41]. The advance of SMRT technology has opened the door for a more comprehensive understanding of AS events. While beyond the scope of this study, the high-confidence curated transcriptome of hSAECs contains >113 K isoforms encoded by 13,038 genes. Predictably, the size distribution of unspliced genes was shorter than that of multi-exonic genes (Figure 2A). Of multi-exonic genes, a remarkable 45% encoded six or more mRNA transcripts. Almost certainly, this number is an underestimate that will be refined as long-read RNA sequencing technology and the informatics tools available to analyze these data mature. Our finding that the vast majority of RNA sequences map to within 50–100 nt of the annotated transcription start sites (Figure 2E) argues that our data are of high quality and the cDNA synthesis is representative of the hSAEC transcriptome. More work will be required to identify, characterize, and validate the novel transcripts in the curated hSAEC transcriptome.

The major focus of this study was to identify the effect of the RSV induced innate response on AS events. Previous work analyzing the global gene expression profile using high-density microarrays [2,16] and short-read RNA sequencing [17] has been published. These studies have been successful in identifying time-dependent, replication-sensitive [18], and NFκB-controlled gene expression programs [16,19] in response to RSV infection. Here, RSV replication can affect the expression of >1000 epithelial genes in a time- and MOI-dependent manner, including up- and downregulation [2]. From these studies, it is well established that the inducible transcriptome encodes innate responses and growth-factor-associated remodeling pathways [17]. Although this gene-level analysis has been informative to elucidate key signaling mediators and characterize products of innate defense, we lack a deep understanding of how mRNA processing controls the diversity of the cell stress response.

### 4.2. mRNA Processing Mediates Cell Stress Responses

This systematic analysis of host mRNA processing in response to virus infection is the first in our knowledge. We note previous work that has shown alternative mRNA processing is implicated in diverse cellular processes, including differentiation [42,43], tissue identity [44], and cellular stress responses [45], so it is not surprising that RSV induces mRNA processing. These earlier studies involving nutrient deprivation found that AS mediates rapid and specific changes in ribosomal subunit splicing and polyadenylation site selection in response to environmental stress responses in yeast [45]. In these systems, alternative mRNA processing impacts gene expression through noncoding RNA interference and altering miRNA regulation. In addition, AS influences protein translation rates and produces proteoforms with acceptors for post-translational modification and/or distinct degradation properties. Our differential polyadenylation analysis indicates that RSV infection induces substantial differential polyadenylation site utilization of ribosomal subunits and proteins important in the dynamics of mRNA processing, protein secretion, and translation. The full biological effects of alternative polyadenylation utilization will require further experimentation.

Remarkably, pathway and network enrichment studies have identified the IIR and cell cycle regulation as being two major biological processes involving mRNA processing and DIU in RSV-infected epithelial cells. While not a focus of this study, RSV infection induces cell cycle arrest through autocrine regulation of TGFβ1 secretion, a phenomenon that is associated with enhanced RSV replication in primary lung epithelial cells [46]. The role of mRNA processing in cell cycle arrest or TGFβ secretion will require further investigation.

A focus of this work, both IFN signaling and IFN regulators, IRF/NFκB are known to be major components of the RSV-inducible genomic response. We and others have shown that RSV replication is a potent inducer of type I and III IFNs, proteins that play important roles in the local antiviral response, immune activation, and cell fate determination [3,12,47]. IFNs upregulate over 350 antiviral genes in adjacent non-infected cells; these proteins serve to induce resistance to viral replication, limiting the spread of virus [48].

### 4.3. Mechanisms of How Transcriptional Initiation Is Coupled to Post-transcriptional mRNA Processing

Although the factors controlling AS in RSV-infected epithelial cells are likely complex and will require additional research, full-length mRNA sequencing studies have linked AS to the control of transcriptional initiation [49]. In this seminal study, Anvar et al. showed a marked interdependency between transcription and mRNA-processing events, including AS and APA, in MCF-7 breast carcinoma cells. Although the factors coupling transcriptional initiation to AS were not investigated in that study, a great deal of understanding has been developed for how transcriptional initiation is linked to mRNA processing in IIR gene expression. Chromatin immunoprecipitation and siRNA depletion studies have shown that the process of transcriptional elongation is responsible for activating the immediate–early IFN response by a chromatin remodeling complex known as positive transcription elongation factor-b (PTEF-b). PTEF-b is a multisubunit protein complex whose core regulators are cyclin-dependent kinase 9 (CDK9) and bromodomain-containing protein 4 (BRD4) recruited to immediate–early innate response genes by the sequence specific binding of IRF and RelA transcription factors [50]. Once recruited to an innate-inducible gene, the PTEF-b complex phosphorylates the carboxy terminal domain of RNA polymerase II, stimulating RNA polymerase II processivity and dissociating transcriptional elongation inhibitors from the promoter [51]. Quantitative proteomics of the effect of innate activation on the CDK9 complex has found that this complex dynamically associates with the DDX5/17 family of RNA helicases. DDX5/17 play a role in co-transcriptional alternative splicing of redox regulatory genes [52], an essential component of the antiviral response to RSV [53]. Collectively, these data indicate that highly inducible immediate–early genes are co-transcriptionally spliced and processed via dynamic changes in the transcriptional elongation complex.

### 4.4. AS of the RelA Transcription Factor

We focused on extensive validation of the RelA and IRF7 transcription factors, master regulators of the IIR [54,55]. Previous mechanistic and single-molecule work has shown that most components of the NFκB pathway are subject to alternative splicing and that these isoforms are induced by pathway activation (reviewed in [56]). For example, an AS product of the IKKγ/NEMO signaling adapter of IKK skips exon 5, known as IKKγ∆. Expression of IKKγ∆ alters the signalsome, enabling it to couple viral replication to NFκB activation, but inhibits IRF3 activation, resulting in an impaired IFN response [12]. With regard to RelA, previous work has shown that mRNA encoding an alternative splice acceptor site located 30 nt into exon 8 encodes p65Δ, a protein lacking a component of the DNA-binding Rel homology domain, required for association with p50 and for DNA binding [57]. Although we do not observe p65∆ in our analysis, p65∆ appears to be a major transcript in pre-B cells and may be developmentally regulated [58]. Our data extend the understanding of inducible mRNA events to include processes of cryptic exon inclusion, exon occlusion, and intron retention of the RelA gene. Our work is foundational for leading to a deeper understanding of the effects of mRNA processing on RelA signaling in the IIR.

### 4.5. Inducible IRF7 mRNA Processing

In this work, we focused on IRF7, a highly NFκB-inducible transactivator of the mucosal type I IFN response [54]. Previous mechanistic work has found that the *irf7* gene encodes four isoforms, IRF7A, IRF7B, IRF7C, and IRF7D (IRF7H) [59]. One mechanistic study examining IRF7 processing described that inducible expression of alternatively spliced IRF7 transcripts with a GC-rich intron and a weak 5′ splice site occurred. This study established alternative splicing decreased mature IRF7 transcripts and protein levels [60]. Here, expression of the intron-retained IRF7-isoform-dampened type I IFN response to single-stranded virus infection. The role of intron retention in controlling the expression of fully spliced gene expression will need to be further developed.

### 4.6. Intron Retention (IR) in the IIR

The rapid response and magnitude of the IIR is a critical factor in the ability of the host to respond to infectious viruses. Through its paracrine actions to activate IFN-stimulated genes, type I IFN expression is critical in reducing the initial spread of viruses. However, prolonged or exuberant IIR activation can be also detrimental, leading to cytokine storm syndrome [61] and cellular apoptosis [15]. The mechanisms how the IIR is terminated is not well understood.

Our study may shed light on how the IIR can be attenuated through mRNA-processing regulators of the IIR. Both RelA and IRF7, master upstream regulators of the IIR, are encoded by genes that express transcripts with retained introns. Next-generation sequencing technologies have enabled the detection of numerous transcripts that retain introns (reviewed in [62]). Although intron retention (IR) typically does not contribute to proteomic diversity, IR events have the ability to act as negative regulators of gene expression. The mechanisms by which IR transcripts function as negative regulators are by slowing down splicing kinetics [63] or increasing potential cytoplasmic degradation by nonsense-mediated decay [62], delaying rapid gene expression.

## 5. Conclusions

In conclusion, we apply SMRT sequencing to understand the role of mRNA processing in the IIR to RSV infection. A high-quality curated transcriptome of hSAECs was generated and alternatively processed isoforms quantitated by short-read RNA sequencing. RSV infection induces differential polyadenylation of transcripts important in protein and mRNA processing, affecting signal recognition, translation, and nonsense mRNA decay pathways. Strikingly, a number of DIU events are observed corresponding to genes controlling the cell cycle and type I IFN pathways. We validated a number of novel isoforms of the master regulators of the IIR, RelA and IRF7. These DIU events primarily include exon inclusion, exon skipping, and intron retention. Our data provide a foundation for a further understanding of the role of mRNA processing in controlling the activation and termination of the IIR to RSV infection.

## Figures and Tables

**Figure 1 viruses-13-00218-f001:**
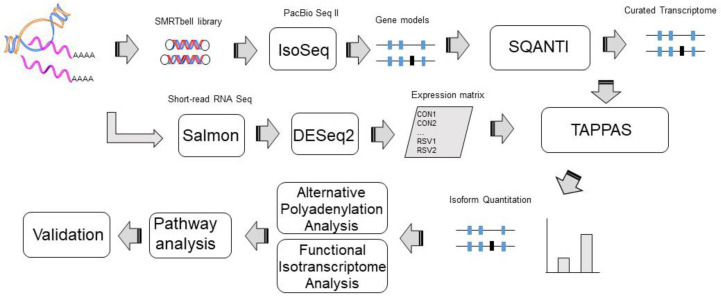
Data analysis pipeline for quantitative mRNA processing. Shown is a schematic representation of the data analysis pipeline for integrating single-molecule, real-time (SMRT) with short-read RNA sequencing analysis. Filtered, polished reads from PacBio Sequel II were pooled from control and respiratory syncytial virus (RSV)-infected hSAECs and used to generate a curated transcriptome, filtering novel not in catalog (NNIC) and RNA transcripts. Short-read RNA transcripts were mapped to hg38 and transcript abundance determined in Salmon 1.3. The curated transcriptome, mRNA expression matrix, and experimental design was used in tappAS to quantify differential isoform utilization (DIU) and analyze differences in annotations.

**Figure 2 viruses-13-00218-f002:**
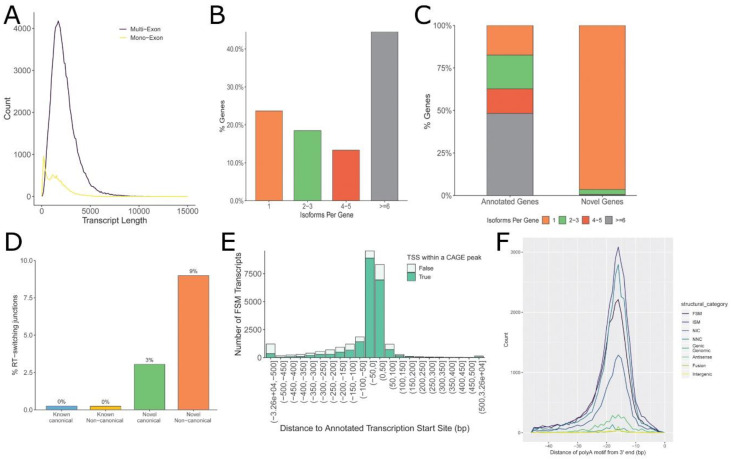
Characteristics of the curated human small-airway epithelial cell (hSAEC) transcriptome. (**A**) Size distribution of transcripts for multi-exon and mono-exon genes. (**B**) Number of isoforms per gene. (**C**) Comparison of isoforms identified for annotated vs. novel genes. (**D**) Percentage of reverse transcriptase (RT) switching. Note that full splice match (FSM) and incomplete splice match (ISM) switching is 0%. (**E**) Distance of annotated reads to known transcription start sites (TSS). Note the high percentage of reads mapped within 20 nt of known TSS, indicating high-quality reverse transcription of the complementary DNA (cDNA) library. (**F**) Distribution of transcripts to known polyadenylation (polyA) sites. The median location is within 18 nt of known polyA sites for all transcript categories.

**Figure 3 viruses-13-00218-f003:**
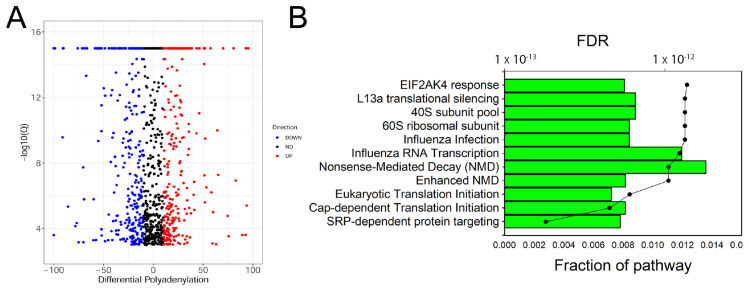
Alternative polyadenylation analysis. (**A**) Scatter plot of differentially polyadenylated transcripts in response to RSV infection. *X* axis: differential polyadenylation utilization (DPAU). DPAU refers to the fraction of transcripts utilizing the distal polyadenylation site. A positive-valued DPAU indicates enhanced distal site polyadenylation in RSV-infected cells, while a negative value indicates distal site utilization in uninfected cells. *Y* axis: −log10 (*Q*-value) of significance of change. (**B**) Pathway enrichment of genes undergoing DPAU. Shown are the top 10 gene pathways (Reactome.org) for a fraction of the pathway represented (boxes) in the query data and the false discovery rate (FDR, circles) of each pathway.

**Figure 4 viruses-13-00218-f004:**
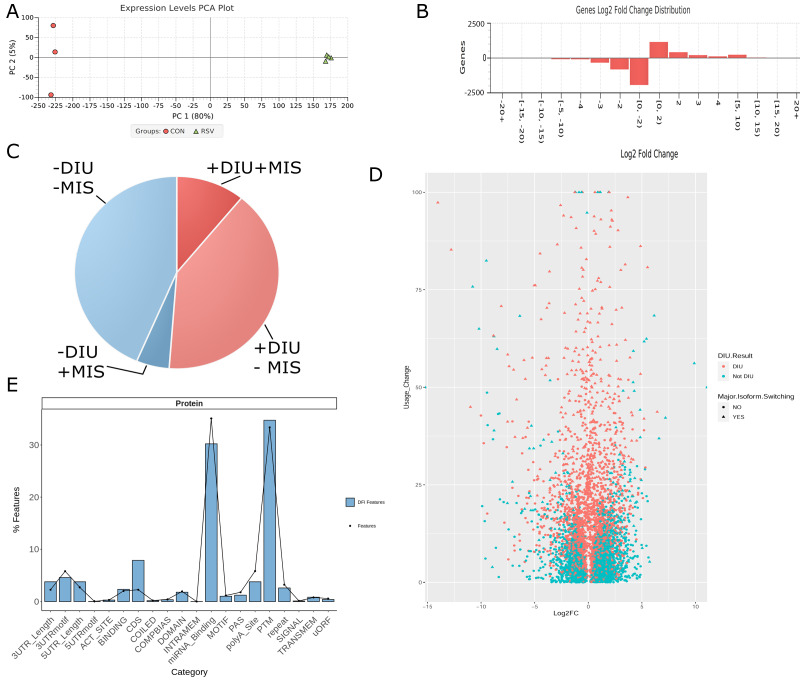
Differential expression analysis. (**A**) Principal component analysis (PCA) of replicate transcriptomes. Note that principal component 1 accounts for >80% of the variance and represents the effect of RSV infection. (**B**) Changes in gene expression for differentially expressed transcripts. (**C**) distribution of differential isoform utilization (DIU) and major isoform shifting (MIS). (**D**) Scatterplot of DIU and MIS in RSV induced differentially expressed transcripts. X axis is log 2 (Fold Change) RSV vs mock infected control. Y axis is Usage Change, an expression of the change in isoforms as a result of RSV infection. (**E**) Functional categories of differential exon splicing in differentially expressed full splice matches (blue boxes) relative to those found in all splice forms (dot symbols). Note that the alternative splicing (AS) exons containing miRNA-binding sites decreased and exons encoding protein domains (coding sequences (CDS)) and exons that are targets for post-translational modifications (PTMs) increased in the RSV-induced DIUs.

**Figure 5 viruses-13-00218-f005:**
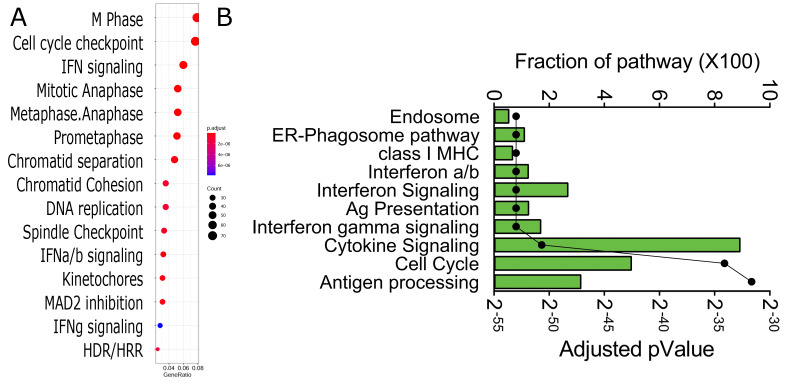
Pathway analysis of differentially expressed DIU transcripts. (**A**) Shown are genome ontologies of enriched DIU transcripts ranked by the number of genes (gene ratio) and by enrichment relative to the genome (pValue). (**B**) Biological pathways of DIUs. For each pathway are shown the fraction of the pathway contained within the DIU transcript list and the adjusted *p* Value.

**Figure 6 viruses-13-00218-f006:**
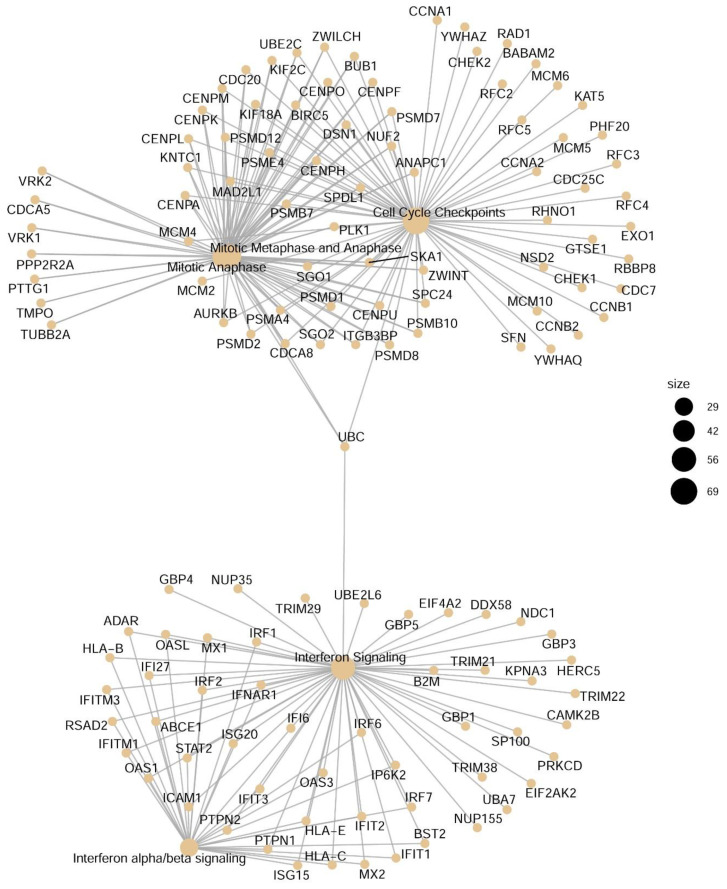
Integrated network analysis of RSV-induced transcripts with differential usage (DIU) were analyzed for network interaction. Node size is indicated in the legend.

**Figure 7 viruses-13-00218-f007:**
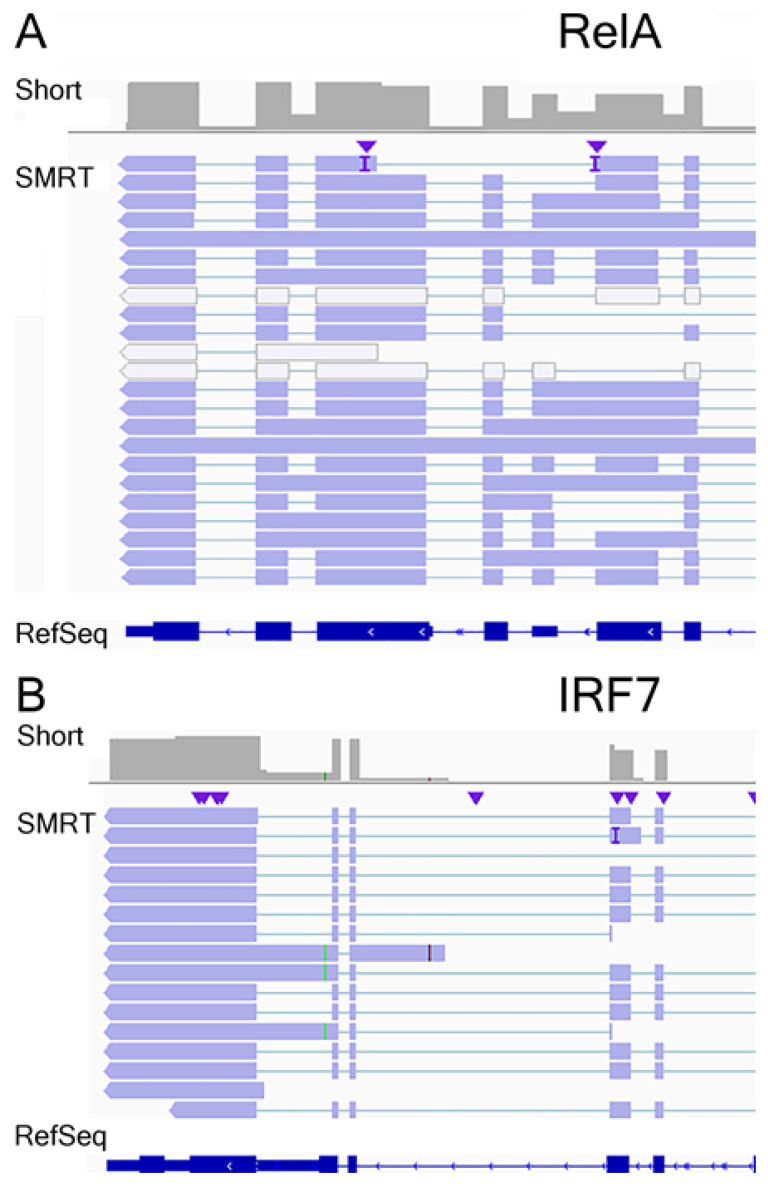
AS of NFκB/RelA and interferon regulatory factor (IRF7) mRNAs. Integrated Genomic Browser view of pile-up of short-read RNA sequencing (Short, in gray) and individual transcripts (SMRT) for (**A**) RelA and (**B**) IRF7. Genome organization detailing intron–exon boundaries are indicated in dark blue at the bottom. The abundance of short-read sequencing is a most reliable estimate of relative expression levels. Note the presence of exon exclusion, exon skipping, and intron inclusion in transcripts quantified by SMRT sequencing.

**Figure 8 viruses-13-00218-f008:**
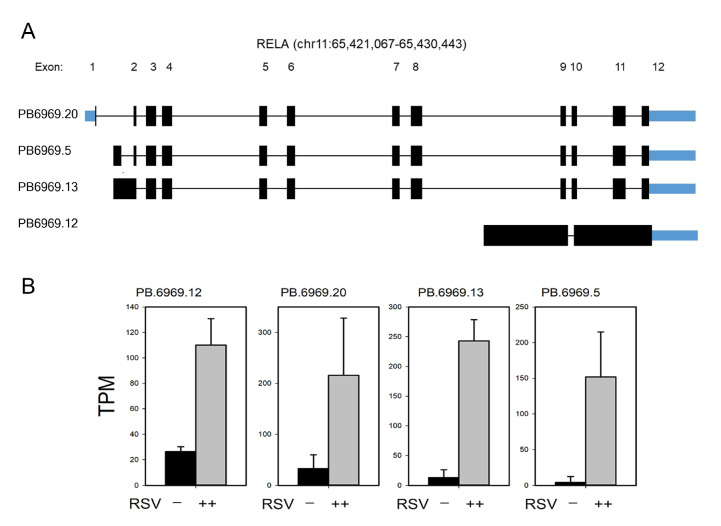
Annotation and expression of NFκB/RelA AS forms. (**A**) Transcript annotation for NFκB/RelA. Exons are numbered at the top, relative to the RefSeq annotation; genomic coordinates are indicated. Coding exons are indicated in black; noncoding exons are indicated in blue. PacBio sequence identifiers are at the left. PB6969.5 is a novel transcript. (**B**) Quantitation of the most abundant RelA transcripts from short-read RNA sequencing. Data are presented as transcripts/million (TPM). Note that RSV infection induces major isoform switching, resulting in DIU.

**Figure 9 viruses-13-00218-f009:**
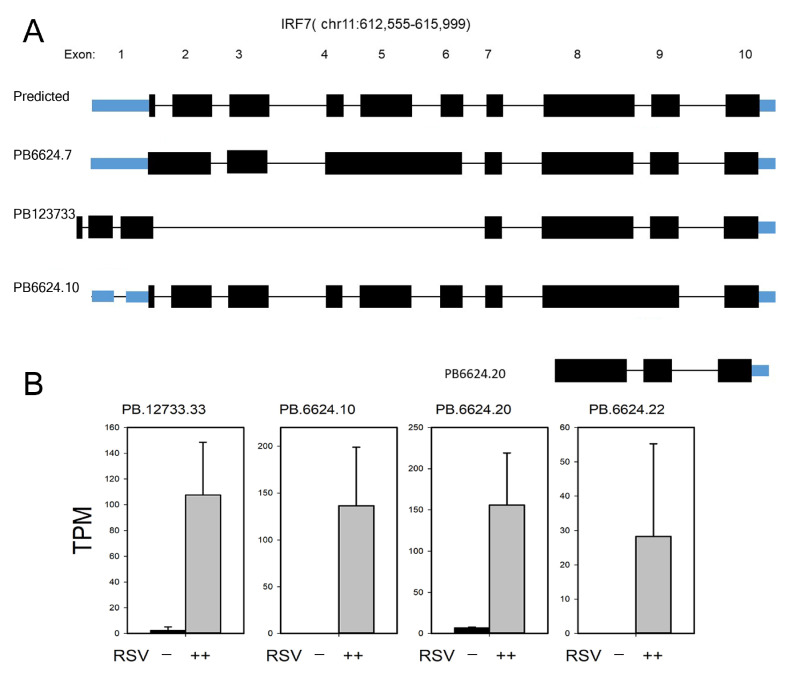
Annotation and expression of IRF7 AS forms. (**A**) Transcript annotation of IRF7. Exons are numbered at the top, relative to the RefSeq annotation; genomic coordinates are indicated. Coding exons are indicated in black; noncoding exons are indicated in blue. PacBio sequence identifiers are at the left. PB123733, PB6624.10, and PB6624.20 are novel transcripts. (**B**) Quantitation of the most abundant IRF7 transcripts from short-read RNA sequencing. Data are presented as transcripts/million (TPM). Note that RSV infection induces major isoform switching.

**Figure 10 viruses-13-00218-f010:**
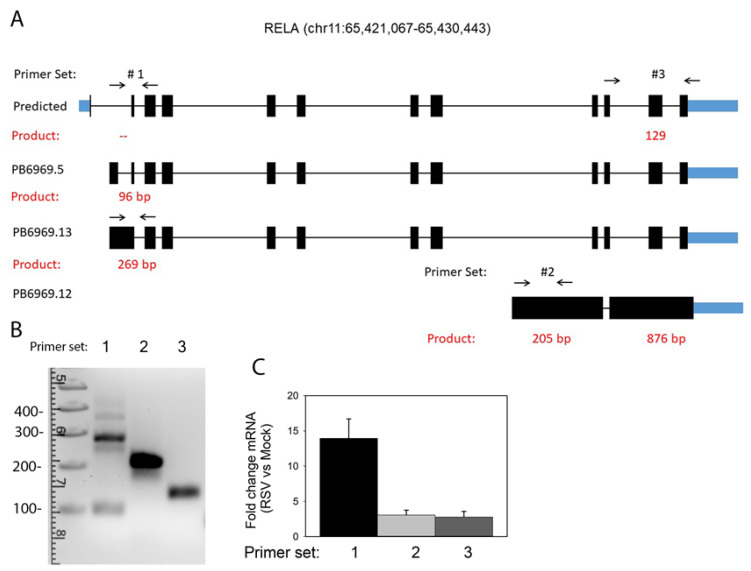
Validation of RelA alternative mRNA processing. (**A**) Primer sets #1-#3 were designed to validate differential exon inclusion/exclusion. The expected size of the PCR product is indicated in red and tabulated in Table II. (**B**) Quantitative PCR for each primer set. hSAECs were infected with RSV and RNA analyzed for isoform expression. Shown is 2% agarose gel electrophoresis stained with GelRed dye (Biotium), with molecular weight (MW, in nt) indicated at the left. Fragments were isolated and sequenced to confirm splice site selection. (**C**) Quantitation of RelA isoforms expressed as a fold change in RSV-infected over uninfected cells (mock).

**Figure 11 viruses-13-00218-f011:**
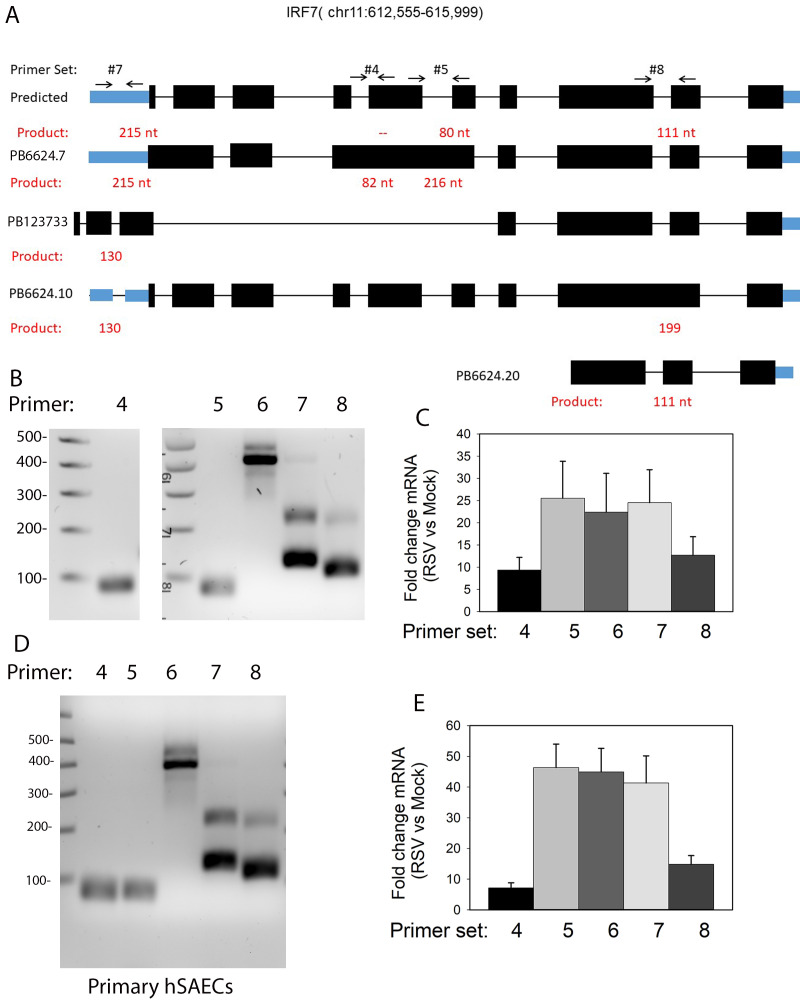
Validation of IRF7 alternative mRNA processing. (**A**) Predicted exons of IRF7. Annotated exons are numbered. For each AS event, primer sets #4-#8 were designed to validate differential exon inclusion/exclusion. The expected size of the PCR product is indicated in red (**B**) Quantitative PCR for each primer set. Shown is agarose gel electrophoresis, with MW (in nt) indicated at the left (**C**) Quantitation of IRF7 transcripts from qRT-PCR. (**D**) Validation of AS in primary hSAECs. Agarose gel electrophoresis as in panel (**C**). (**E**) Quantitation of IRF7 transcripts from qRT-PCR findings indicate that RSV-induced mRNA processing of the IIR is a general response of the small-airway epithelium.

**Table 1 viruses-13-00218-t001:** Isoform-selective PCR primer sets.

Target Gene (Pair)	Forward Primer (5′–3′)	Reverse Primer	Target Transcript	Size (nt)
*RelA 1*	ATTCCTTACCCCGTTTTCCCTC	AATGATCTCCACATAGGGGCCA	PB6969.5	96
			PB6969.13	269
			RelA-WT	None
*RelA 2*	CATGTGGTCTTAGAGGGGCAG	GAGAAGTGGGACTTGCTCTCC	PB6969.12	205
			RelA-WT	None
*RelA 3*	TCCTTTCAGCGGACCCAC	TTGATGGTGCTCAGGGATGAC	PB6969.12	876
			RelA-WT	129
*IRF7 4*	GCTGCTTACGTTCACCCTGAC	CTTGGAGTCCAGCATGTGTGTG	PB6624.7	82
			IRF7-WT	None
*IRF7 5*	AAGAAGGGCTTCCCCTGACT	CTCTACTGCCCACCCGTACA	PB6624.7	216
			IRF7-WT	80
*IRF7 6*	GAGGCCCGCACCTGTTT	TTGGGGAGGGTGACAGGTA	PB123733	123
			IRF7	637/724
*IRF7 7*	AGCCCTTACCTCCCCTGTTA	GCTGATCTCTCCAAGGAGCC	PB6624.10/PB123733	130
			IRF7-WT	215
*IRF7-8*	CTCGGAACTGTGACACCCCC	AGCCCAGGTAGATGGTATAGCG	PB6624.10	199
			IRF7-WT	111

**Table 2 viruses-13-00218-t002:** Characterization of transcripts based on splice junctions. Abbreviations: FSM, full splice match; ISM, incomplete splice match; NIC, novel in catalog; NNC, novel not in catalog.

Category	Isoform (No.)
FSM	28,561
ISM	35,495
NIC	14,972
NNC	28,166

## Data Availability

The data presented in this study are openly available in GEO (accession number GSE161846; https://www.ncbi.nlm.nih.gov/geo/query/acc.cgi?acc=GSE161846).

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
