# Peer review of "Alternative mRNA Processing of Innate Response Pathways in Respiratory Syncytial Virus (RSV) Infection"

_viruses, 2021, doi:10.3390/v13020218_

Round 1

Reviewer 1 Report

The authors have produced a high quality manuscript based on the use of state-of-the-art molecular biology techniques. Their study has improved the understanding at the post-transcriptional level of the innate immune response to RVS.

Minor remarks:

  • What is the genetic background of the cell hSAECs strain used in this study? Is it possible that it has been previously exposed to the RSV?

  • In the methodology, the authors do not indicate how the hSAECs were infected with the RSV and the time required after infection to conduct the transcriptomic study.
    Also, did the authors use a negative control (hSAECs not infected with the virus)?

Author Response

Thank you for taking the time to review our manuscript.

The genotyping and phenotypic characterization of the hSAECs has been published.  These cells were isolated from a cadaveric donor, 22 year old male.  Cytogenetic analysis and array comparative genomic hybridization profiling (in Reference # 21, Ramirez et al) show immortalized HBECs to have duplication of parts of chromosomes 5 and 20, but otherwise demonstrate stable epithelial differentiation markers over many population doublings. Its more than likely that the person was infected with RSV in the past, but there is no evidence of RSV genome expression in by RT-PCR or in our proteomic studies.  We cite the reference #21 in the methods for readers to consult if interested.

Methods of infection- on lines 104 in Methods Section we state:  "Virus was adsorbed onto the surface of hSAECs in submerged culture for 2 h prior to washing at an MOI of 1.  Mock infected cells were used as control. "

Reviewer 2 Report

Respiratory syncytial virus (RSV) is a leading cause of respiratory illness in infants. Even though transcriptome studies using RSV-infected airways epithelial cells have been done in the past, studies like the manuscript by Xiaofang Xu and colleagues reveal the need to investigate in a more detail manner the complexity of post-transcriptional events such as complex alternate splicing and polyadenylation that are involved in the control of the innate immune response to RSV infection. The authors also addressed that this study is only the basis to understand the effects of mRNA processing on signaling the IRR.

The content of the manuscript is presented in a clear and organized style that makes it easy for the reader to follow and understand the points that the authors are trying to address. There is also a good use of the literature available to discuss the different sections of the manuscript.

Here are my suggested revisions:

Line 59: Need to add citation.

Line 115 and 120: The core facility where the sequencing was done. Was the same RNA used for both sequencing technologies?

Line 182: For this section, I would like to see a supplementary table showing which genes are the ones the authors make reference and if they were up or downregulated. Which genes are associated with the pathways shown in your graphic?

Line 204: Same as comment above. Could you expand more information on what are those genes which ones are up/down regulated? Be more specific and show more data.

Figure 7: This figure is very busy and some of the texts are almost on top of each other. I will suggest breaking the figure maybe in two or adding a supplementary figure. 

Does the short reads height represent the number of reads aligning to a given gene? Could explain more in detail what we are looking at?

 Figure 8A and B: I will suggest adding the RefSeq and labeling in more detail what the reader is looking at. There are symbols that are not clear what they are describing.

Figures 9 and 10: The numbers of the exons are confusing. I will suggest adding the RefSeq. What blue black, thin lines means.

Author Response

thank you for reviewing our paper and the constructive comments to improve its quality.

  1. We have cited the work showing that the inflammasome coordinates activation of both NFkB and IRF via the IkB kinase nemo adapter.
  2. The same RNA was used for making both short read and SMRT libraries.  We have clarified this point in the methods section
  3. We have included 4 additional supplementary tables referred to in the text.  Supplemental Table I is the complete data set for the 905 transcripts undergoing differential polyadenylation with their expression levels and confidence scores in control and RSV infected state.  Supplemental Table II is the reactome pathways of the APA, with enrichment score and individual genes that were mapped to these pathways. Supplemental Table III is the individual genes and enrichment scores for the transcripts demonstrating differential isoform utilization.  Supplemental Table IV is the individual genes, and their expression levels for the significantly enriched pathways.  These data are also described in the Results section of the text.
  4. We have replotted figure 7, removing IFNL, IFIT and IRF1, to illustrate the alternative splicing of IRF7 and RelA, the major points of this paper.  the important message of this figures is that the short read RNA seq -given by the grey bars- give independent confirmation at the exon-level of the alternative splicing events.  
  5. Figure 8, we have removed the confusing TappAS representation of coding and noncoding exons and replaced this with our RefSeq annotations and exon numbering. Blue exons are noncoding exons; black are coding.  These are clarified in the figure legends.
  6. We have simplified figures 9-11 to more clearly illustrate the primer design for confirmation of the AS events.  

Round 2

Reviewer 2 Report

here are my suggestions:

1. I would suggest to deposit the Raw and processed RNA-Seq data in NCBI's Gene Expression Omnibus to make it accessible for future use.

2. Figures 8 and 9: Which of the isoforms are the news ones and which ones were already in the NCBI database?. Also, could the authors make it clear in figure 8A that you are only showing the most abundant AS and not the 5 REIA isoforms. 

1. Add title table 2.

Author Response

1.  Raw and processed RNA seq is deposited in GEO under accession number GSE161846, indicated in Data Availability.

2. In figure legend 8A, we indicate PB6969.5 is a novel transcript.  In Figure 8B, we indicate this is quantitation of the most abundant RelA transcripts from short read RNA-Seq. 

In figure legend 9A, we indicate PB123733 and PB6624.10 and PB6624.20 are novel transcripts. In Figure 9B, we indicate this is quantitation of the most abundant IRF7 transcripts from short read RNA-Seq. 

3. Table II title has been added " Isoform selective PCR primer sets"